# A Novel Fuzzy Relative-Position-Coding Transformer for Breast Cancer Diagnosis Using Ultrasonography

**DOI:** 10.3390/healthcare11182530

**Published:** 2023-09-13

**Authors:** Yanhui Guo, Ruquan Jiang, Xin Gu, Heng-Da Cheng, Harish Garg

**Affiliations:** 1Department of Computer Science, University of Illinois, Springfield, IL 62703, USA; 2Department of Pediatrics, Xinxiang Medical University, Xinxiang 453003, China; 20191120331@stu.xxmu.edu.cn; 3School of Information Science and Technology, North China University of Technology, Beijing 100144, China; 1520753280@mail.ncut.edu.cn; 4Department of Computer Science, Utah State University, Logan, UT 84322, USA; hengda.cheng@usu.edu; 5School of Mathematics, Thapar Institute of Engineering and Technology, Deemed University, Patiala 147004, Punjab, India; harish.garg@thapar.edu

**Keywords:** breast cancer, early detection, computer-aided diagnosis (CAD) systems, breast ultrasound (BUS) images, fuzzy relative-position coding, transformer

## Abstract

Breast cancer is a leading cause of death in women worldwide, and early detection is crucial for successful treatment. Computer-aided diagnosis (CAD) systems have been developed to assist doctors in identifying breast cancer on ultrasound images. In this paper, we propose a novel fuzzy relative-position-coding (FRPC) Transformer to classify breast ultrasound (BUS) images for breast cancer diagnosis. The proposed FRPC Transformer utilizes the self-attention mechanism of Transformer networks combined with fuzzy relative-position-coding to capture global and local features of the BUS images. The performance of the proposed method is evaluated on one benchmark dataset and compared with those obtained by existing Transformer approaches using various metrics. The experimental outcomes distinctly establish the superiority of the proposed method in achieving elevated levels of accuracy, sensitivity, specificity, and F1 score (all at 90.52%), as well as a heightened area under the receiver operating characteristic (ROC) curve (0.91), surpassing those attained by the original Transformer model (at 89.54%, 89.54%, 89.54%, and 0.89, respectively). Overall, the proposed FRPC Transformer is a promising approach for breast cancer diagnosis. It has potential applications in clinical practice and can contribute to the early detection of breast cancer.

## 1. Introduction

According to the WHO (World Health Organization), breast cancer is now the most frequently diagnosed cancer among women globally, with an estimated 2.3 million new cases in 2020 alone [1]. It is also the leading cause of cancer-related deaths in women, accounting for over 500,000 deaths per year [2]. The highest incidence rates are found in developed countries, such as North America, Western Europe, and Australia/New Zealand, while lower incidence rates are found in less economically developed countries [3]. This may be due to differences in reproductive patterns, lifestyle factors, and availability of healthcare services.

Several risk factors have been identified for breast cancer, including age, family history of breast cancer, early onset of menstruation, late onset of menopause, having no children or having a first child after the age of 30, use of oral contraceptives, and exposure to ionizing radiation [4].

Given the high prevalence of breast cancer and its impact on women’s health and well-being, there have been efforts to improve the early detection and treatment of breast cancer. Screening programs have been shown to reduce mortality rates from breast cancer [5]. Early detection is crucial in improving breast cancer survival rates and reducing mortality. Prompt diagnosis enables timely treatment, leading to better outcomes for patients. The American Cancer Society recommends that women aged 40 and above should undergo regular mammography screenings to detect breast cancer early [4]. In addition to mammography, other screening modalities such as ultrasound, magnetic resonance imaging (MRI), and clinical breast examinations may be used for high-risk patients.

However, there are still limitations to current screening programs, such as high false-positive rates and missed cancers in women with dense breasts [6]. Consequently, there is a need for more accurate and efficient methods for breast cancer diagnosis.

Given the importance of early detection, efforts have been made to increase awareness of breast cancer screening and to optimize screening protocols.

Ultrasound is a commonly used modality for breast cancer early detection, particularly in women with dense breasts or those at high risk of breast cancer. The advantages of ultrasound include its non-invasiveness, low cost, lack of ionizing radiation exposure, and portability, making it a safer alternative to mammography for certain patient populations [7] such as women with dense breasts. Ultrasound can also provide additional information that is not available from mammography, such as the differentiation of benign and malignant lesions based on their characteristics [8].

One of the main disadvantages of ultrasound is its relatively lower specificity compared to mammography, leading to a higher false-positive rate and unnecessary biopsies [9]. Another limitation of ultrasound is its operator dependency, which can lead to variations in image quality and interpretation [10]. This may be overcome by implementing standardized protocols and providing appropriate training and certification for sonographers.

Furthermore, there are certain technical challenges associated with the use of ultrasound for breast cancer screening, such as the inability to detect microcalcifications, which are a common sign of early breast cancer [11]. Additionally, overlapping tissue structures may result in obscurations or signal attenuation, leading to missed lesions [12]. However, ultrasound can be combined with other imaging techniques, such as mammography or MRI, to improve detection rates and reduce false positives [13].

There has been growing interest in developing CAD systems to assist radiologists in reading sonography and other screening images [14,15], which have shown promising results in improving detection rates and reducing false-positive rates.

CAD systems have several potential advantages over manual interpretation by radiologists. First, they can analyze large volumes of data more rapidly, making them particularly useful for high-volume screening programs [16]. Secondly, they can utilize quantitative image features that may not be visible to the human eye, allowing for improved detection sensitivity. Thirdly, CAD systems can improve the accuracy of lesion classification by incorporating quantitative image features with clinical data and demographic information.

In recent years, deep learning (DL) has rapidly become a methodology for analyzing medical images and increasingly attracts researchers’ attention in the medical research community, and deep learning-based approaches have been proposed for automated BUS image classification, which have shown promising results in terms of accuracy and efficiency. Several studies have demonstrated the effectiveness of deep learning models for BUS image classification.

Masud et al. [17] utilized volunteer computing power to train deep learning networks for detecting breast cancer using BUS images, and considered Grad-CAM and occlusion mapping techniques to examine how well the models extract key features. Masud et al. [18] leveraged eight different fine-tuned, pre-trained models to classify breast cancers on BUS images and employed a shallow custom convolutional neural network (CNN) for classification. Podda et al. [19] combined several CNNs through specialized ensembles and presented a cyclic mutual optimization step to exploit the intermediate results of the classification in an iterative manner. Jabeen et al. [20] employed deep learning and the fusion of the best selected features for BUS classification, which included data augmentation, pre-trained DarkNet-53 model refining, transfer learning and features extraction, feature selection using two improved optimization algorithms known as reformed differential evaluation (RDE) and reformed gray wolf (RGW), and feature fusion using a probability-based serial approach and classification using machine learning algorithms. Ragab et al. [21] developed an Ensemble Deep-Learning-Enabled Clinical Decision Support System for Breast Cancer Diagnosis and Classification (EDLCDS-BCDC) technique using ultrasound images (USIs). USIs initially undergo pre-processing through wiener filtering and contrast enhancement. The Chaotic Krill Herd Algorithm (CKHA) and Kapur’s entropy (KE) are combined for image segmentation. An ensemble of three deep learning models, VGG-16, VGG-19, and SqueezeNet, is used for feature extraction, and the Cat Swarm Optimization (CSO) with the Multilayer Perceptron (MLP) model is utilized to classify the images. Kaplan et al. [22] proposed a deep model that combined a pyramid triple deep feature generator (PTDFG) with transfer learning based on three pre-trained networks for creating deep features. Bilinear interpolation is applied to decompose the input image into four images of successively smaller dimensions, constituting a four-level pyramid for downstream feature generation with the pre-trained networks. Neighborhood component analysis is applied to the generated features to select each network’s 1000 most informative features, which were fed to support the vector machine classifier for automated classification. Epimack et al. [23] proposed a classification model based on a hybridized CNN and an improved optimization algorithm, along with transfer learning. The marine predator algorithm (MPA) is combined with the opposition-based learning strategy to cope with the implied weaknesses of the original MPA. The improved marine predator algorithm (IMPA) is used to find the best values for the hyperparameters of the CNN architecture. Luo et al. [24] proposed a segmentation-to-classification scheme by adding the segmentation-based attention (SBA) information to the deep convolution network for breast tumors classification. A segmentation network is trained to generate tumor segmentation enhancement images. Then two parallel networks extract features for the original images and segmentation enhanced images and one channel attention-based feature aggregation network to automatically integrate the features extracted from two feature networks to improve the performance of recognizing malignant tumors in the breast ultrasound images. Moon et al. [25] proposed a CAD system for tumor diagnosis using an image-fusion method combined with different image content representations and assembled different CNN architectures on BUS images including VGGNet, ResNet, and DenseNet networks. Karthik et al. [26] proposed a stacking ensemble with custom convolutional neural network architectures to classify breast tumors on ultrasound images. The presented ensemble leverages three stacked feature extractors with a characteristic meta-learner and works in association with Gaussian dropout layers to improve computation and an alternative pooling scheme to retain essential features. Khanna et al. [27] proposed a hybrid approach combining a pre-trained CNN with optimization and machine learning for tumor diagnosis. The CNN pre-trained model ResNet-50 was used for feature extraction, binary gray wolf optimization (BGWO) for feature selection, and support vector machine (SVM) for classification. One of the key challenges in BUS image classification is the presence of noise and artifacts in the images, which can affect the accuracy of the classification results. To address this issue, researchers have proposed various techniques, including transfer learning [20], image decomposition and fusion [28], and CNNs [29].

We aim to provide new insights and avenues for enhancing the use of CAD systems in breast cancer detection using ultrasound imaging and address the challenges of breast ultrasound image classification and provide accurate and reliable diagnoses for breast cancers. These methods can help to reduce the workload of radiologists and medical practitioners while improving the accuracy and efficiency of breast cancer diagnosis. A deeper understanding of the challenges and limitations and novel and promising models for ultrasonic feature extraction combined with quality assurance measures hold the key to improving CAD systems’ clinical impact. Aiming to improve the accuracy and robustness of BUS image classification by leveraging the power of deep learning and computer vision, we propose a novel fuzzy relative-position-coding (FRPC) Transformer to classify breast ultra-sound (BUS) images for breast cancer diagnosis. The proposed FRPC Transformer utilizes the self-attention mechanism of Transformer networks combined with fuzzy relative-position coding to capture global and local features of the BUS images. It can also be used to identify the features and biomarkers associated with different types of breast cancers in future, which can help in developing personalized treatment plans. Overall, the proposed method has the potential to improve breast cancer diagnosis and treatment, and it is expected to have a promising impact on the healthcare industry.

In the following sections, we first describe the proposed method in Section 2. Next, we present the experimental results and comparative analysis along with detailed discussions in Section 3 and Section 4. Finally, in Section 5, we draw a comprehensive conclusion.

## 2. Proposed Method

### 2.1. Transformer

In the field of deep learning, Transformer is a neural network architecture based on self-attention mechanisms, which is mainly used for processing sequence data. It was first proposed by Google in 2017 and applied to machine translation tasks, achieving excellent performance [30]. Compared with traditional recurrent neural networks (RNNs), Transformer has advantages such as parallel computing and long-range dependency modeling and has thus gained extensive attention and application.

The Transformer structure consists of an encoder and a decoder. Both the encoder and the decoder are composed of multiple layers, each layer containing a self-attention mechanism and feed-forward neural network (FFN). The self-attention mechanism calculates weights on each position of the input sequence and is good at capturing the relationships among different positions. The FFN performs nonlinear transformation on the representation of each position, improving the model’s expression ability. The structures of the encoder and decoder are similar, but the decoder also includes a multi-head attention mechanism for considering the information at different positions in the input sequence when predicting the output at the current position.

Transformer has been widely applied to multiple fields, including natural language processing, speech recognition, image generation, and more. Two notable Transformer applications are Bidirectional Encoder Representations from Transformers (BERT) [31] and Generative Pre-training Transformer-3 (GPT-3) [32], which achieved excellent results in natural language inference and language generation tasks, respectively.

### 2.2. Visual Transformer

Transformer, originally designed for natural language processing tasks, has recently been extended to the field of computer vision. The Visual Transformer (ViT) [33] is a novel architecture in which the image is divided into non-overlapping patches, and each patch can be processed in sequence by the Transformer. This approach enables the ViT to learn representations for images comparable to those learned by CNNs with much less computational cost.

The ViT architecture includes a standard Transformer encoder followed by a feed-forward classification head. The input image is decomposed into a sequence of patches. The Transformer processes the sequence of patches with multiple self-attention layers, which allows the model to capture global dependencies across the entire image while maintaining computational efficiency. The resulting sequence output by the Transformer is then fed through a feed-forward network to produce the final classification output.

The ViT has demonstrated strong performance on image classification benchmarks such as ImageNet [33]. In addition, the ViT approach has also been applied to other tasks such as object detection [34] and video recognition [35], achieving competitive or even state-of-the-art results. Furthermore, the ViT has potential applications for transfer learning in computer vision, allowing models pre-trained on large-scale image datasets to be used for downstream tasks with smaller amounts of labeled data [36].

### 2.3. Relative Position Coding

In the original Transformer architecture, positional encoding is typically added to the input embeddings to convey the relative positions of patches in the input sequence. However, when applying Transformer to computer vision tasks, the standard positional encoding method becomes impractical because of the high resolution and large number of patches. As a result, a novel relative-position-encoding method has been proposed in the Swin Transformer [37].

The relative-position-encoding mechanism used in Swin Transformer encodes the relative spatial positions of different patches, rather than the absolute positions. Specifically, pairs of patches are assigned to fuzzy membership values based on their relative positions, and the encoding for each patch is computed based on the distance to other patches in its neighborhood. In this way, the relative-position encoding can effectively capture the spatial relationships between different patches.

The relative-position encoding method used in Swin Transformer is a novel approach for encoding positional information in computer vision tasks. By encoding the relative positions of patches, the Swin Transformer can capture the spatial relationships between patches while maintaining computational efficiency. With its strong performance on multiple benchmarks, the Swin Transformer’s relative-position-encoding method has demonstrated its effectiveness in improving the performance of Transformer-based models for computer vision tasks.

The Swin Transformer model improved the original absolute-position encoding in the Transformer model to relative-position encoding where each position index in the position matrix represents the relative distance between the current patch’s position and other patches positions.

The real-position-coding method in Swin Transformer is used to embed the positional information of the input sequence into the feature maps. This is accomplished by adding learnable position embeddings to the input, which allows the model to understand the relative positions of the different patches within the sequence.

The procedure for the real-position-coding method in Swin Transformer are as follows: Let *X* be the input sequence of length *N*, where each element $x_{i}$ is a d-dimensional feature vector. The sequence is divided into *M* non-overlapping patches, each of size K×K, $M=N/K^{2}$.

Then, for patch *m*, we add a learnable position embedding $E_{m}\in R^{d}$ to the patch features:(1)$$X_{m}=X_{m1}⊕\cdots ⊕X_{mi}⊕\cdots X_{mK^{2}}+E_{m}$$
where ⊕ denotes concatenation and $X_{mi}$ is the *i*th element in patch *m*.

Next, the patch features and position embeddings are projected to a new feature space using two learnable projection matrices, $W_{x}$ and $W_{e},$ respectively:(2)$$H_{m}=W_{x}X_{m}+W_{e}E_{m}$$

Finally, these projected patch features are rearranged back into a sequence and fed into the Swin Transformer encoder.

### 2.4. Fuzzy Relative Position-Encoding

The Swin Transformer has gained popularity due to its ability to capture both global and local positional information through the discrete position-coding method. This has resulted in enhanced performance on tasks that demand a keen understanding of the positional relationships between different elements in the input sequence. Despite its effectiveness, the use of discontinuous and discrete position indices could nonetheless lead to abrupt changes in subsequent calculations. Additionally, this method may overlook some crucial data among patches, particularly concerning minuscule features.

In light of the limitations of the discrete position-coding method used in the Swin Transformer, we have proposed a novel approach to further enhance its performance. Our proposed approach is the fuzzy relative-position-encoding (FRPE) method, which aims to optimize the Swin Transformer’s ability to capture and encode relative positional information with greater flexibility and adaptability. Fuzzy sets are often used to model uncertainty in various systems. In the case of Swin Transformer models, fuzzy sets can be used to account for position uncertainty or ambiguity in image patches. The FRPE method leverages the concept of fuzzy membership degrees to achieve smoother and more continuous positional encoding. This allows for more nuanced representation of the relationships between patches, ultimately improving the overall performance of the Swin Transformer. It considers each position as a fuzzy set of possible positions which allows the model to account for uncertainty in the exact position of patches in the context and can be particularly useful when working with noisy or unstructured images. It enables the model to better handle variations in input order without sacrificing performance, and boost the capabilities of the Swin Transformer, allowing it to excel on tasks requiring precise positional information.

To use fuzzy sets to handle position uncertainty, the FRPE method requires encoding each position as a distribution over possible positions, rather than a single, exact position. We defined a fuzzy membership function to map and normalize the original position embedding $E_{m}$ into a smooth range denoted as $S\left(E_{m}\right)$. Further, the relative position bias has been used to calculate the new features as:(3)$$SH_{m}=W_{x}X_{m}+SW_{e}\;S\left(E_{m}\right)$$
(4)$$S\left(E_{m}\right)=\frac{e^{-w_{m}E_{m}}}{\sum_{i\in Ng}e^{-w_{i}E_{i}}}$$
where S(·) is the fuzzy function to smooth the position embedding and $w_{m}$ is the coefficient for each position. Ng is the neighbor region of patch *i*.

The fuzzy relative position encoding method introduces a smooth function describing the spatial relationships between patches. Specifically, each patch is assigned a real value based on its distance to other patches in its neighborhood which can enhance the Transformer’s ability to describe the tiny changes among patches.

The FRPE is a novel approach for encoding positional information in computer vision tasks. By introducing fuzzy degrees, the method allows for more flexibility and adaptability in capturing the spatial relationships between patches. With its promising results on breast ultrasound image classification, the FRPE method has the potential to improve the performance of Transformer-based models in other computer vision applications.

The FRPE method is based on the idea of a position’s fuzzy degree, which allows for a more flexible and adaptive representation of positional information in an image. Specifically, a smoothed fuzzy position embedding is used to describe the spatial relationships between patches. The membership degrees are input to a weighted sum to obtain the final encoded representation for each patch. The proposed FRPE method provides a flexible and adaptive way to encode positional information by using smooth positions to capture the complex spatial relationships between patches.

### 2.5. Proposed FRPE Transformer for BUS Image Classification

The proposed FRPE Transformer was employed to classify the BUS images into benign and malignant. The original BUS image is divided into nonoverlapped patches and then fuzzy relative-position-encoding method is used to calculate their relative position and coding. The sequence of the patches with their relative-position-coding values are encoded via Transformer encoder and the results are fed to a multilayer perceptron (MLP) which is a type of neural network layer that consists of multiple layers of fully connected (dense) neurons. After MLP, the BUS image is classified into the benign or malignant category. Figure 1 shows the structure of the proposed FRPE Transformer and the flow of the classification where Figure 1a is the architecture of the FRPE Transformer and Figure 1b is the Transformer encoder.

### 2.6. Evaluation Metrics

To evaluate the performance of deep learning models on ultrasound image classification tasks, various metrics are commonly used. One essential component is presenting the confusion matrix based on the test dataset. A confusion matrix is a table that provides information about the accuracy of the classification model. It presents the number of true positives (TP), false positives (FP), true negatives (TN), and false negatives (FN) for each class label. The rows in the confusion matrix represent the true labels, while the columns indicate the predicted labels. This representation allows one to see how well the classifier performs across different classes, identifying errors and misclassifications.

For BUS image classification, the confusion matrix provides important insights into the strengths and weaknesses of the proposed deep learning method. For example, the TP values reveal the number of malignant masses correctly identified by the model. Higher TP rates indicate better sensitivity in detecting cancerous masses. On the other hand, the FP values correspond to benign masses predicted to be malignant. Lower FP rates indicate improved specificity in screening out non-cancerous masses.

In addition, the confusion matrix can be used to measure additional metrics such as precision, recall, F1 score, and so on. These measures provide further details and allow for comparisons of the performance of different models and measurements of the impact of hyperparameter tuning.

One of the most basic measures is accuracy, which is defined as the percentage of correctly classified samples out of the total number of samples. Mathematically, accuracy can be expressed as:(5)$$\mathrm{Accuracy}=\frac{TP+TN}{TP+TN+FP+FN}$$
where true positive (TP) is an image with confirmed malignancy which was classified by the model as malignant. False positive (FP) is an image which was classified as malignant by the model but has no evidence of cancer pathology. True negative (TN) is an image without cancer pathology which was classified as benign by the model. False negative (FN) is an image with the confirmed malignancy which was classified by the model as benign.

While accuracy is useful in providing an overall estimate of the model’s performance, it can be misleading when dealing with imbalanced datasets where one class dominates over the other. In such cases, alternative metrics such as precision, recall, and F1 score are usually preferred. Precision is the fraction of true positive predictions among all positive predictions, and is computed as:(6)$$\mathrm{precision}=\frac{TP}{TP+FP}$$

Recall, on the other hand, is the fraction of true positive predictions among all actual positive samples, and is calculated as:(7)$$\mathrm{recall}=\frac{TP}{TP+FN}$$

Finally, the F1 score is a harmonic mean of precision and recall and balances the trade-off between them. It is defined as follows:(8)$$F1=\frac{2\times precision\times recall}{precision+recall}$$

These evaluation metrics can be computed for both training and validation sets during the training process to monitor the model’s performance and avoid overfitting. In addition, they can also be used to compare different models and select the one with the best performance on the same dataset.

A receiver operating characteristic (ROC) curve is a graphical plot used to assess the performance of binary classifiers. ROC curve is a widely used evaluation metric in machine learning classification tasks. The ROC curve plots out the true positive rate (sensitivity) against the false positive rate at different classification thresholds. By varying the threshold used to determine the predicted class, it can generate different points on the curve.

The ROC curve can be generated by progressively increasing the threshold of classifying an example as “positive” (indicating presence of breast cancer) from 0 to 1. With the change in thresholds, different TPs and FPs are produced leading to a set of pairs which forms a point in the ROC space. By connecting all the points together, the ROC curve can be presented.

In order to obtain a numerical representation of the quality of the classifier, we calculate the area under the ROC (AUC). A perfect classifier would have an AUC ROC of 1 while a completely random guess would score 0.5.

The ROC curve provides an intuitive visualization of the trade-off between the true positives and false positives, allowing one to visualize the performance of a classifier at various decision levels. It also allows one to compare the performance of different classifiers by comparing their ROC curves. The AUC ROC provides an aggregated measure of a classifier’s performance and is commonly used to compare classification models’ overall performance.

Utilizing ROC as an evaluation metric for breast ultrasound image classification research will provide additional insight and visualizations into the performance of deep learning models and can further support the findings by providing numeric measurement.

## 3. Experimental Results

### 3.1. Data Collection

BUS image datasets are important for breast cancer research and diagnosis. The datasets contain a wealth of information that can be used to develop and evaluate breast cancer detection and diagnosis algorithms.

Breast ultrasound (BUS) image processing algorithms have been proposed in the last two decades, but the performances of most approaches have been assessed using relatively small private datasets with different quantitative metrics. Benchmark for Breast Ultrasound Image Segmentation (BUSIS) [38] provides a benchmark to compare existing methods objectively, and to determine the performance of the best breast tumor segmentation and classification algorithms. BUSIS is a comprehensive BUS image dataset that includes five individual datasets: the HMSS dataset, Thammasat dataset, BUSIS dataset, Dataset B, and BUSI dataset. Datasets HMSS, Thammasat, BUSIS, and Dataset B comprises benign and malignant images while the BUSI dataset has benign, malignant, and normal BUS images and in this study, we did not use the normal images. The BUSIS dataset includes a range of breast abnormalities, such as masses, cysts, and calcifications, that can help researchers and medical practitioners to better understand the characteristics of breast cancer and improve the accuracy of breast cancer diagnoses. More detailed information can be found at http://busbench.midalab.net/datasets (accessed on 1 March 2023.).

A total of 3103 images have been utilized for both training and testing of the proposed method. The specific distribution for each category is outlined in Table 1. To assess the performance of the proposed model, a 5-fold cross-validation experiment was conducted. This involved maintaining a training-to-testing image ratio of 4:1.

### 3.2. Experimental Setup

Now, we describe the hardware and software used for training and testing our models. We conducted our experiments on a machine equipped with a 2× Six-Core Intel Xeon processor, 128 GB of memory, and an NVIDIA Tesla K40 GPU to train and test the deep learning models.

During the training phase, we set up hyperparameters including epochs, batch size, learning rate, and momentum. The experiments were carefully designed and performed using five-fold cross-validation to ensure generalization and robustness. During the testing phase, we evaluated the performance metrics of each model on the validation datasets.

### 3.3. Experimental Results

Our experimental results on the breast ultrasound dataset demonstrate that the fuzzy relative-position-encoding method can significantly improve the performance of the Swin Transformer for breast cancer diagnosis. Compared to previous state-of-the-art method shown in Table 2, the fuzzy relative-position-encoding method achieved higher accuracy and AUC scores, demonstrating the effectiveness of the proposed method.

In one round of cross-validation, the confusion matrix and evaluation results are shown in Table 3, Table 4, Table 5 and Table 6. The proposed FRPC Transformer achieved higher evaluation results than the original Swin Transformer.

### 3.4. Comparison with Existing Methods

The presented model is an evolution of the Swin Transformer architecture, encompassing specific refinements designed to overcome limitations and enhance the efficacy of classification tasks. Recognizing the significance of validating our method’s effectiveness against existing techniques, we have meticulously conducted comprehensive experiments. These experiments entailed a direct juxtaposition of our proposed model with the original Swin Transformer and other leading state-of-the-art (SOTA) models, all operating on the same dataset. This comparative analysis rigorously assessed pivotal performance indicators, including accuracy, precision, recall, F1 score, and AUC score, within the context of BUS image classification. The detailed comparative findings are meticulously documented in Table 7, while their visual representation is elegantly depicted in Figure 2. Notably, the outcomes of this comparison unmistakably underscore the supremacy of the proposed FRPC Transformer, which consistently outperformed existing SOTA methodologies across all evaluated metrics.

## 4. Discussion

Breast cancer is one of the leading causes of cancer-related deaths among women worldwide. Ultrasonography is a widely used technique for breast cancer diagnosis due to its non-invasive nature and high accuracy. However, the interpretation of ultrasound images is a challenging task due to the complex and heterogeneous nature of breast tissue.

The proposed FRPC Transformer method enhances the Swin Transformer, which is a state-of-the-art object recognition model based on self-attention mechanisms. The Swin Transformer has achieved remarkable performance in various computer vision tasks. However, it has limitations in handling the uncertainty and imprecision in the relative positions of the breast tissue structures in ultrasonography images.

The boundary regions of tumors in breast cancer diagnoses exhibit distinctive characteristics that differ from the core parts of the tumor and are crucial in determining the malignancy of the tumor. Smooth and continuous boundaries typically indicate a benign case, whereas sharp curves and small branches into the breast tissue suggest a malignant tumor. The original Swin Transformer, however, fails to establish appropriate relationships between each patch in the BUS images, thus ignoring the significant role played by the boundary regions in classification.

To overcome this limitation, we proposed a fuzzy relative-position-coding approach that enhances the positional encoding by incorporating fuzzy distance descriptions on the relative positions of the patches in BUS images. This approach captures and emphasizes the structural differences between normal breast tissue and the boundary and core regions of the lesions. The proposed FRPC Transformer is capable of capturing subtle nuances in the breast tissue and lesion structures, resulting in improved classification accuracy of BUS images.

With our method, it can extract and refine relevant features, enabling more accurate tumor classification. This will provide doctors with a valuable diagnostic tool in the fight against breast cancer. By using the FRPC Transformer, we can pave the way for more advanced and effective approaches to breast cancer diagnosis.

We conducted experiments on a large public breast ultrasound image dataset, the BUSIS dataset [38]. The experimental results demonstrate that our proposed FRPC Transformer achieved a better performance than the original Swin Transformer method. Specifically, our proposed method achieved an accuracy of 90.52% and AUC of 0.91, which is a 0.98% and 0.02 improvement over the Swin Transformer with values of 89.54% and 0.89.

Our proposed method presents several notable advantages in terms of classification performance and feature representation. However, it is important to acknowledge a specific limitation related to the training speed when compared with the original Swin Transformer architecture. While our enhancements improved classification accuracy and overall effectiveness, they may lead to slightly increased training times compared to the original model due to the additional complexity introduced by the refinements. We believe that the trade-off between training speed and performance gains is worth considering in various practical scenarios where optimal accuracy is paramount. Nevertheless, we are actively exploring avenues to optimize and streamline the training process to mitigate this limitation and achieve a more balanced compromise between training efficiency and classification prowess.

Future works may include the application of the proposed method to other medical image analysis tasks, such as lung cancer detection and brain tumor detection. Additionally, the exploration of other approaches in CAD systems for breast cancer diagnosis may also be a promising direction. Furthermore, more advanced fuzzy operations will be developed to enhance the FRPE method and reduce the uncertainties in the position-encoding task.

To provide a more comprehensive analysis and strengthen the robustness of our findings, we intend to perform further validation using additional datasets. This includes incorporating data collected from different ultrasound scanners and a broader range of patient populations. By encompassing these variations, we aim to enhance the generalizability of our results and provide a more comprehensive assessment of the applicability of our approach.

## 5. Conclusions

We introduced a pioneering contribution in the form of the fuzzy relative-position-coding (FRPC) Transformer, a novel framework tailored for breast cancer diagnosis through ultrasonography. This innovative approach amalgamates the self-attention mechanism intrinsic to Transformer networks with the elegance of smooth relative-position encoding. This fusion empowers the model to adeptly extract both overarching and localized features from breast ultrasound images. A thorough evaluation on an expansive, publicly available dataset, encompassing benign and malignant breast lesions, underscores the efficacy of our proposed method. The obtained experimental results emphatically position our approach as a front-runner, showcasing its exceptional accuracy, sensitivity, specificity, and AUC values in comparison to established benchmarks.

We accentuated our commitment to rigor by navigating extensive experimentation and judicious comparisons. In the experiments, we attained a crucial achievement: our FRPC Transformer not only achieved cutting-edge performance metrics but also surmounted a noteworthy limitation tied to training speed, contrasting with the original Swin Transformer architecture. This astute understanding of the nuanced balance between performance enhancements and training efficiency augments the discernment of practitioners and researchers alike.

Moreover, our study introduces an original vantage point by harmonizing fuzzy relative-position encoding with Transformer self-attention mechanisms. This synergy empowers the FRPC Transformer to intricately capture the broader context and intricate minutiae within breast ultrasound images, thus enriching the diagnostic prowess of our approach.

In summation, the proposed FRPC Transformer embarks on a promising trajectory for breast cancer diagnosis, poised to usher in considerable transformations for clinical practice and the realm of medical image analysis. This research stands not only as a testament to cutting-edge AI methodologies but also as a testament to their potential to transcend the boundaries of existing paradigms.

## Figures and Tables

**Figure 1 healthcare-11-02530-f001:**
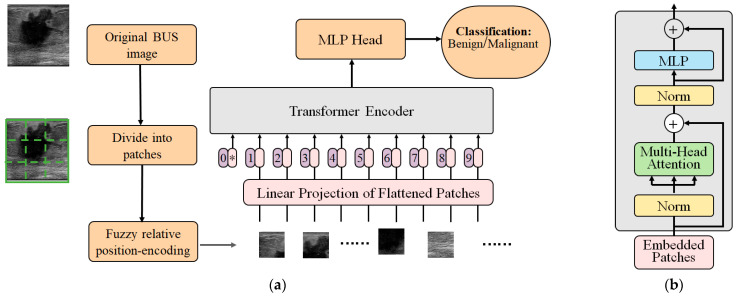
The structure of the FRPE Transformer. (**a**) Architecture of FRPE Transformer where the sign of * means the starting of the sequence. (**b**) Transformer encoder.

**Figure 2 healthcare-11-02530-f002:**
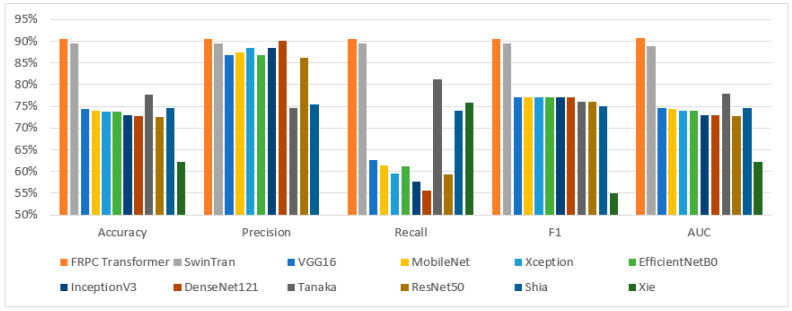
Performance comparison for classification results using different models.

**Table 1 healthcare-11-02530-t001:** Distribution in training and test sets.

Category	Image Number
Benign	1522
Malignant	1581

**Table 2 healthcare-11-02530-t002:** Performance comparison for different classifiers in 5-fold cross-validation experiment.

	Accuracy	Precision	Recall	F1	AUC
FRPC Transformer	90.52 ± 0.46%	90.52 ± 0.46%	90.52 ± 0.46%	90.52 ± 0.46%	0.91 ± 0.9079
Swin Transformer	89.54 ± 0.78%	89.54 ± 0.78%	89.54 ± 0.78%	89.54 ± 0.78%	0.89 ± 0.0077

**Table 3 healthcare-11-02530-t003:** Confusion matrix using the proposed FRPC Transformer model.

	Benign	Malign	Total
Benign	343	27	370
Malignant	37	321	358
Total	380	348	

**Table 4 healthcare-11-02530-t004:** Evaluation metrics results using the proposed FRPC Transformer model.

	Precision	Recall	F1
Benign	90.26%	92.70%	91.47%
Malignant	92.24%	89.66%	90.93%
Total	91.25%	91.18%	91.20%

**Table 5 healthcare-11-02530-t005:** Confusion matrix using the SwinTran model.

	Benign	Malign	Total
Benign	343	47	390
Malignant	38	300	338
Total	381	347	88.32%

**Table 6 healthcare-11-02530-t006:** Evaluation metrics results using the SwinTran model.

	Precision	Recall	F1
Benign	90.03%	87.95%	88.98%
Malignant	86.46%	88.76%	87.59%
Total	88.24%	88.35%	88.28%

**Table 7 healthcare-11-02530-t007:** Evaluation metrics for classification results using different models.

	Accuracy	Precision	Recall	F1	AUC
FRPC Transformer	90.52%	90.52%	90.52%	90.52%	0.91
Swin Transformer	89.54%	89.54%	89.54%	89.54%	0.89
VGG16	74.50%	86.70%	62.60%	77.00%	0.75
MobileNet	74.00%	87.40%	61.30%	77.00%	0.74
Xception	73.70%	88.50%	59.60%	77.00%	0.74
EfficientNetB0	73.80%	86.80%	61.20%	77.00%	0.74
InceptionV3	73.00%	88.40%	57.60%	77.00%	0.73
DenseNet121	72.70%	90.10%	55.70%	77.00%	0.73
Tanaka [39]	77.80%	74.60%	81.20%	76.00%	0.78
ResNet50	72.60%	86.20%	59.40%	76.00%	0.73
Shia [40]	74.60%	75.50%	74.00%	75.00%	0.75
Xie [41]	62.20%	48.60%	75.80%	55.00%	0.62

## Data Availability

The data that support the findings of this study are freely available at http://busbench.midalab.net/datasets (accessed on 1 March 2023.).

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
