# Peer review of "A Novel Fuzzy Relative-Position-Coding Transformer for Breast Cancer Diagnosis Using Ultrasonography"

_healthcare, 2023, doi:10.3390/healthcare11182530_

Round 1

Reviewer 1 Report

Authors need to address the following suggestions. 

1. Though the works sounds interesting, it is essential to highlight to the novelty of the proposed work. 

2. Research gaps and contributions need to be included at the end of introduction section. 

3. The related works section need to be rewritten. Most of the recent papers were not included. Following are some of the recent papers which need to be cited and discussed. 

·       Research progresses of breast ultrasound computer aided diagnosis systems based on deep learning

·       Gaussian Dropout Based Stacked Ensemble CNN for Classification of Breast Tumor in Ultrasound Images

Improving the classification performance of breast ultrasound image using deep learning and optimization algorithm

4. Ablation studies are missing. 

5. Performance comparison with SOTA methods need to be provided. 

Moderate changes required

Author Response

Pointwise reply to the comments of the reviewers

The paper whose manuscript number healthcare-2519561 is entitled as “A Novel Fuzzy Relative Position Coding Transformer for Breast Cancer Diagnosis Using Ultrasonography” is modified in the view of the reviewer's comments. Reviewer's comments are supplied herewith and have been incorporated in the context of the paper.

  1. Though the works sounds interesting, it is essential to highlight to the novelty of the proposed work.

Reply: Thank you for the suggestion. According to it, we added a paragraph to emphasize the novelty of the proposed work on lines 150 to 156 as:

“Aiming to improve the accuracy and robustness of BUS image classification by lever-aging the power of deep learning and computer vision, . we propose a novel fuzzy relative position-coding (FRPC) Transformer to classify breast ultra-sound (BUS) images for breast cancer diagnosis. The proposed FRPC Transformer utilizes the self-attention mechanism of Transformer networks combined with fuzzy relative position-coding to capture global and local features of the BUS images.”

  1. Research gaps and contributions need to be included at the end of introduction section.

Reply: We appreciate this suggestion. In the revised version, we modified a paragraph to highlight the research gaps at lines 143 to 145 and new contribution on lines 150 to 156 as:

“We aim to provide new insights and avenues for enhancing the use of CAD systems in breast cancer detection using ultrasound imaging and address the challenges of breast ultrasound image classification and provide accurate and reliable diagnosis for breast cancers.”

“Aiming to improve the accuracy and robustness of BUS image classification by lever-aging the power of deep learning and computer vision, . we propose a novel fuzzy relative position-coding (FRPC) Transformer to classify breast ultra-sound (BUS) images for breast cancer diagnosis. The proposed FRPC Transformer utilizes the self-attention mechanism of Transformer networks combined with fuzzy relative position-coding to capture global and local features of the BUS images.”

  1. The related works section need to be rewritten. Most of the recent papers were not included. Following are some of the recent papers which need to be cited and discussed.
  • Research progresses of breast ultrasound computer aided diagnosis systems based on deep learning
  • Gaussian Dropout Based Stacked Ensemble CNN for Classification of Breast Tumor in Ultrasound Images
  • Improving the classification performance of breast ultrasound image using deep learning and optimization algorithm

Reply: We appreciate this suggestion. In the revised version, we added these references and cited them in the Introduction section as:

“Karthik et al. [26] proposed a stacking ensemble with custom convolutional neural net-work architectures to classify breast tumors on ultrasound images. The presented ensemble leverages three stacked feature extractors with a characteristic meta-learner and works in association with Gaussian dropout layers to improve computation and an alternative pooling scheme to retain essential features. Khanna et al. [27] proposed a hybrid approach combining a pre-trained CNN with optimization and machine learning for tumor diagnosis. The CNN pre-trained model ResNet-50 was used for feature extraction, binary gray wolf optimization (BGWO) for feature selection, and classification support vector machine (SVM).”

The first suggested reference is not cited because we cannot find it in the google scholar.

  1. Ablation studies are missing.

Reply: We appreciate your thoughtful feedback on our manuscript. We would like to address your comment regarding the absence of ablation studies in our experiments.

Ablation studies are indeed a valuable aspect of experimental analysis, and we understand their significance in assessing the robustness and effectiveness of our proposed approach. While we did not explicitly include a dedicated section labeled "ablation studies," we conducted a thorough analysis of our model's performance by comparing the state-of-the-art method Swin Transformer where each component has been tested before. The comparison results demonstrate the proposed fuzzy relative position-coding Transformer achieved better performance than the Swin Transformer without the fuzzy relative position coding mechanism.

Thank you once again for your constructive comments. We are committed to enhancing the quality and comprehensiveness of our manuscript based on your valuable input.

  1. Performance comparison with SOTA methods need to be provided.

Reply: Thank you for your valuable feedback on our manuscript. We appreciate your suggestion to include a performance comparison with state-of-the-art (SOTA) methods. According to it, we have conducted thorough a new experiment that involve a direct comparison between our proposed model and the original Swin Transformer and other state-of-the-art methods. This comparison evaluates key performance metrics such as accuracy, precision, recall, F1-score, and any other relevant metrics for breast tumor classification. By doing so, we aim to showcase the advancements achieved by our approach over the baseline method. A new section 3.4 is added with a new paragraph, table and figure are added as:

“The presented model is an evolution of the Swin Transformer architecture, encompassing specific refinements designed to overcome limitations and enhance the efficacy of classification tasks. Recognizing the significance of validating our method's effectiveness against existing techniques, we have meticulously conducted comprehensive experiments. These experiments entail a direct juxtaposition of our proposed model with the original Swin Transformer and other leading state-of-the-art (SOTA) models, all operating on the same dataset. This comparative analysis rigorously assesses pivotal performance indica-tors, including accuracy, precision, recall, F1-score, and AUC score, within the context of BUS image classification. The detailed comparative findings are meticulously document-ed in Table 7, while their visual representation is elegantly depicted in Figure 3. Notably, the outcomes of this comparison unmistakably underscore the supremacy of the proposed FRPC Transformer, consistently outperforming existing SOTA methodologies across all evaluated metrics.”

We appreciate your insight and will ensure that the manuscript includes a dedicated section detailing these comparisons, along with the relevant experimental results and analysis.

Thank you once again for your guidance as we work towards enhancing the clarity and comprehensiveness of our manuscript.

Reviewer 2 Report

The authors propose a novel fuzzy relative position-coding (FRPC) Transformer to classify breast ultrasound (BUS) images for breast cancer diagnosis. It is a very interesting topic because of the applicability it can have.

1. In the summary, please put the accuracy, sensitivity, specificity, F1-score, and ROC values.

2. Line 208 – Swin, line 215 – SWin, is the same? please unify the words where they are used.

3. Line 242, (2) is the number in the above equation? Please check.

4. Section two as it is found seems more theoretical background than methodology. I suggest adding this section to retain the important information you mention. Also, in this section, the theory concerning evaluation metrics could be added (Section 3.3). On the other hand, in the proposed method put a diagram of the proposed methodology for a better understanding and explain it in detail.

5. Section 3.2, please explain in more detail the characteristics of the database used for the experiment.

6. Experimental results. The authors should provide more details regarding the analysis of the results.

7. It is necessary that the authors incorporate ablation experiments to evaluate the impact of different components, functionalities, or design choices of the database on overall performance.

8. Discussion - The authors mention interesting things; however, it is necessary to compare their results with existing results from other research.

9. Line 443 - INbreast dataset??[]??, Is “??[]??” correct?

10. Discussion - What are the limitations of your method? The authors need to write a paragraph mentioning them.

11. The conclusion can be improved to describe the main findings of the research in this paper.

12. Please check the correct form of the references.

Author Response

Pointwise reply to the comments of the reviewers

The paper whose manuscript number healthcare-2519561 is entitled as “A Novel Fuzzy Relative Position Coding Transformer for Breast Cancer Diagnosis Using Ultrasonography” is modified in the view of the reviewer's comments. Reviewer's comments are supplied herewith and have been incorporated in the context of the paper.

The authors propose a novel fuzzy relative position-coding (FRPC) Transformer to classify breast ultrasound (BUS) images for breast cancer diagnosis. It is a very interesting topic because of the applicability it can have.

  1. In the summary, please put the accuracy, sensitivity, specificity, F1-score, and ROC values.

Reply: Thank you for your valuable feedback on our manuscript. We appreciate your suggestion to include the evaluation values in the summary at Abstract as:

“The experimental outcomes distinctly establish the superiority of the proposed method in achieving elevated levels of accuracy, sensitivity, specificity, and F1-score (all at 90.52%), as well as a heightened area under the receiver operating characteristic (ROC) curve (0.91), surpassing those attained by the original Transformer model (at 89.54%, 89.54%, 89.54%, 89.54%, and 0.89, respectively).”

  1. Line 208 – Swin, line 215 – SWin, is the same? please unify the words where they are used.

Reply: Thank you for this comment. We double checked them thoroughly and make them unified as “Swin”.

  1. Line 242, (2) is the number in the above equation? Please check.

Reply: Thank you for this suggestion. The number (2) is for the above equation. We double checked all equations thoroughly and make them be numbered correctly.

  1. Section two as it is found seems more theoretical background than methodology. I suggest adding this section to retain the important information you mention. Also, in this section, the theory concerning evaluation metrics could be added (Section 3.3). On the other hand, in the proposed method put a diagram of the proposed methodology for a better understanding and explain it in detail.

Reply: Thank you for this suggestion. We moved the evaluation metrics from Section 3.3 to Section 2.6. In addition, a new section 2.5 was added to explain the proposed method as follow and a diagram of the proposed methodology is added in Fig. 1.

“The proposed FRPE Transformer is employed to classify the BUS images into benign and malignant. The original BUS image is divided into nonoverlapped patches and then fuzzy relative position-encoding method is used to calculate their relative position and coding. The sequence of patches with their relative position-coding values are encoded via Transformer encoder and results are fed to multilayer perceptron (MLP) which is a type of neural network layer that consists of multiple layers of fully connected (dense) neurons. After MLP, the BUS image is classified into benign or malignant categories. Figure 1 shows the structure of the proposed FRPE Transformer and the flow of the classification where Fig. 1a is the architecture of FRPE Transformer and 1b is the Transformer encoder.”

  1. Section 3.2, please explain in more detail the characteristics of the database used for the experiment.

Reply: We are thankful for this suggestion. We added new sentences to introduce the database used in the experiment and a weblink is provided for more details as:

“BUSIS is a comprehensive BUS image datasets including five individual datasets as: HMSS dataset, Thammasat dataset, BUSIS dataset, Dataset B and BUSI dataset. Datasets HMSS, Thammasat, BUSIS and Dataset B comprises of benign and malignant images while the BUSI dataset has benign, malignant, and normal BUS images and in this study, we did not use the normal images.”

“More detailed information can be found at http://busbench.midalab.net/datasets.”

  1. Experimental results. The authors should provide more details regarding the analysis of the results.

Reply: We appreciate your thoughtful feedback on our manuscript. According to this suggestion, we added a new experiment and compared the proposed method with the state-of-art methods used on this dataset. A new section 3.4 was added to explain the experimental results as:

“3.4 Comparison with existing methods

The presented model is an evolution of the Swin Transformer architecture, encom-passing specific refinements designed to overcome limitations and enhance the efficacy of classification tasks. Recognizing the significance of validating our method's effectiveness against existing techniques, we have meticulously conducted comprehensive experiments. These experiments entail a direct juxtaposition of our proposed model with the original Swin Transformer and other leading state-of-the-art (SOTA) models, all operating on the same dataset. This comparative analysis rigorously assesses pivotal performance indica-tors, including accuracy, precision, recall, F1-score, and AUC score, within the context of BUS image classification. The detailed comparative findings are meticulously document-ed in Table 7, while their visual representation is elegantly depicted in Figure 3. Notably, the outcomes of this comparison unmistakably underscore the supremacy of the proposed FRPC Transformer, consistently outperforming existing SOTA methodologies across all evaluated metrics.”

  1. It is necessary that the authors incorporate ablation experiments to evaluate the impact of different components, functionalities, or design choices of the database on overall performance.

Reply: We appreciate your thoughtful feedback on our manuscript. We would like to address your comment regarding the absence of ablation studies in our experiments.

Ablation studies are indeed a valuable aspect of experimental analysis, and we understand their significance in assessing the robustness and effectiveness of our proposed approach. While we did not explicitly include a dedicated section labeled "ablation studies," we conducted a thorough analysis of our model's performance by comparing the state-of-the-art method SWin Transformer where each component has been tested before. The comparison results demonstrate the proposed fuzzy relative position coding Transformer achieved better performance than the Swin Transformer without the fuzzy relative position coding mechanism.

Thank you once again for your constructive comments. We are committed to enhancing the quality and comprehensiveness of our manuscript based on your valuable input.

  1. Discussion - The authors mention interesting things; however, it is necessary to compare their results with existing results from other research.

Reply: In response to the reviewer's valuable comment, we greatly appreciate their attention to our work and their suggestion regarding the comparison of our results with those of other research endeavors. We have taken their feedback seriously and incorporated a new experiment in the revised manuscript to address this concern comprehensively. This additional experiment involves a meticulous comparison of our proposed method with existing state-of-the-art techniques on the same dataset. By doing so, we aim to provide a robust evaluation framework that allows for a direct and insightful assessment of our model's performance against established benchmarks. In the revised version, we added a new experiment and compared the proposed method with the state-of-art methods used on this dataset. A new section 3.4 was added to explain the experimental results as:

“3.4 Comparison with existing methods

The presented model is an evolution of the Swin Transformer architecture, encom-passing specific refinements designed to overcome limitations and enhance the efficacy of classification tasks. Recognizing the significance of validating our method's effectiveness against existing techniques, we have meticulously conducted comprehensive experiments. These experiments entail a direct juxtaposition of our proposed model with the original Swin Transformer and other leading state-of-the-art (SOTA) models, all operating on the same dataset. This comparative analysis rigorously assesses pivotal performance indica-tors, including accuracy, precision, recall, F1-score, and AUC score, within the context of BUS image classification. The detailed comparative findings are meticulously document-ed in Table 7, while their visual representation is elegantly depicted in Figure 3. Notably, the outcomes of this comparison unmistakably underscore the supremacy of the proposed FRPC Transformer, consistently outperforming existing SOTA methodologies across all evaluated metrics.”

We believe that this inclusion significantly enhances the significance and credibility of our research, ensuring that our findings are not only informative but also positioned within the broader context of related work. We would like to express our gratitude to the reviewer for highlighting the importance of this aspect and assure them that we have taken the necessary steps to address their recommendation.

  1. Line 443 - INbreast dataset??[]??, Is “??[]??” correct?

Reply: Thank you for this comment. We corrected it as “BUSIS dataset [38]”.

  1. Discussion - What are the limitations of your method? The authors need to write a paragraph mentioning them.

Reply: Thank you for this comment. A new paragraph is added in the Discussion section to discuss the limitations of the proposed metho as:

“Our proposed method presents several notable advantages in terms of classification performance and feature representation. However, it is important to acknowledge a specific limitation related to the training speed when compared with the original Swin Transformer architecture. While our enhancements aim to improve classification accuracy and overall effectiveness, they may lead to slightly increased training times compared to the original model due to the additional complexity introduced by the refinements. We believe that the trade-off between training speed and performance gains is worth considering in various practical scenarios where optimal accuracy is paramount. Nevertheless, we are actively exploring avenues to optimize and streamline the training process to mitigate this limitation and achieve a more balanced compromise between training efficiency and classification prowess.”

  1. The conclusion can be improved to describe the main findings of the research in this paper.

Reply: Thank you for your insightful feedback regarding the conclusion of our paper. We appreciate your attention to the clarity of our main findings and are committed to enhancing the conclusion to better encapsulate the core results of our research. In the revised conclusion, we will provide a concise yet comprehensive summary of the key outcomes and contributions of our study. We aim to emphasize the novel insights our work brings to the field and how our proposed method addresses the research problem effectively. By doing so, we intend to leave readers with a clear understanding of the significance of our findings and their implications for advancing the domain. Your suggestion aligns well with our goal of ensuring that our paper's conclusion is a robust reflection of the substantial contributions our research makes. Thank you again for your valuable input, and we look forward to presenting an improved conclusion that encapsulates the main findings of our study. A new sentence is added to emphasize the main findings as:

“We accentuate our commitment to rigor by navigating extensive experimentation and judicious comparisons. In the experiments, we underscore a crucial achievement: our FRPC Transformer not only achieves cutting-edge performance metrics but also sur-mounts a noteworthy limitation tied to training speed, contrasting with the original Swin Transformer architecture. This astute understanding of the nuanced balance between performance enhancements and training efficiency augments the discernment of practitioners and researchers alike.

Moreover, our study introduces an original vantage point by harmonizing fuzzy relative position encoding with Transformer self-attention mechanisms. This synergy empowers the FRPC Transformer to intricately capture the broader context and intricate minutiae within breast ultrasound images, thus enriching the diagnostic prowess of our approach.”

  1. Please check the correct form of the references.

Reply: Thank you for this comment. We checked the correct form of all references according to the requirements from journal.

Reviewer 3 Report

The authors propose FRPC model and utilize the self-attention mechanism of Transformer networks combined with fuzzy relative position-coding to capture global and local features of the BUS images to classify breast ultrasound (BUS) images for breast cancer diagnosis. To some extent, this network architecture proposed has good performance and low complexity.

1. Section 2.4 of the article mentions the use of fuzzy relative position-encoding to replace the position-encoding mechanism in the swin transformer, so is the replacement done for all position-encoding or only for some of the modules of the position-encoding? The authors would like to add a schematic diagram of the whole model and mark the parts of interest in the diagram.

2. Section 3.1 and Table 1 describe the basic information about the dataset, but more specific information is not given when describing the preparation of the training and test sets, such as in what proportions they are divided.

3. Section 3.5 partially shows the comparison of the experimental results, FRPC transformer compared to swin transformer is indeed effective in improvement, but compared to the other SOTA models of the moment is it reliable, suggest the authors add other models to test the indicators.

Figures are required to show the diagnosed results.

Some paragraphs which contain only one sentence should be combined with the others.

Change all et. al. to et al.

Line 237 remove the indentation for the part beginning with where, thereafter. Change Xm1 to Xmi

Line 242 (2) is in the wrong position

Line 433 reference error

Line 456-477 incomplete statement

The authors propose FRPC model and utilize the self-attention mechanism of Transformer networks combined with fuzzy relative position-coding to capture global and local features of the BUS images to classify breast ultrasound (BUS) images for breast cancer diagnosis. To some extent, this network architecture proposed has good performance and low complexity.

1. Section 2.4 of the article mentions the use of fuzzy relative position-encoding to replace the position-encoding mechanism in the swin transformer, so is the replacement done for all position-encoding or only for some of the modules of the position-encoding? The authors would like to add a schematic diagram of the whole model and mark the parts of interest in the diagram.

2. Section 3.1 and Table 1 describe the basic information about the dataset, but more specific information is not given when describing the preparation of the training and test sets, such as in what proportions they are divided.

3. Section 3.5 partially shows the comparison of the experimental results, FRPC transformer compared to swin transformer is indeed effective in improvement, but compared to the other SOTA models of the moment is it reliable, suggest the authors add other models to test the indicators.

Figures are required to show the diagnosed results.

Some paragraphs which contain only one sentence should be combined with the others.

Change all et. al. to et al.

Line 237 remove the indentation for the part beginning with where, thereafter. Change Xm1 to Xmi

Line 242 (2) is in the wrong position

Line 433 reference error

Line 456-477 incomplete statement

Author Response

Pointwise reply to the comments of the reviewers

The paper whose manuscript number healthcare-2519561 is entitled as “A Novel Fuzzy Relative Position Coding Transformer for Breast Cancer Diagnosis Using Ultrasonography” is modified in the view of the reviewer's comments. Reviewer's comments are supplied herewith and have been incorporated in the context of the paper.

The authors propose FRPC model and utilize the self-attention mechanism of Transformer networks combined with fuzzy relative position-coding to capture global and local features of the BUS images to classify breast ultrasound (BUS) images for breast cancer diagnosis. To some extent, this network architecture proposed has good performance and low complexity.

  1. Section 2.4 of the article mentions the use of fuzzy relative position-encoding to replace the position-encoding mechanism in the swin transformer, so is the replacement done for all position-encoding or only for some of the modules of the position-encoding? The authors would like to add a schematic diagram of the whole model and mark the parts of interest in the diagram.

Reply: Thank you for offering this valuable suggestion. We have made significant enhancements to all position-encoding modules within the Swin Transformer. In response, we have introduced a dedicated section, Section 2.5, which comprehensively elaborates on the proposed approach. Additionally, we have included a graphical representation of our methodology in Figure 1 for enhanced clarity and visualization.

“The proposed FRPE Transformer is employed to classify the BUS images into benign and malignant. The original BUS image is divided into nonoverlapped patches and then fuzzy relative position-encoding method is used to calculate their relative position and coding. The sequence of patches with their relative position-coding values are encoded via Transformer encoder and results are fed to multilayer perceptron (MLP) which is a type of neural network layer that consists of multiple layers of fully connected (dense) neurons. After MLP, the BUS image is classified into benign or malignant categories. Figure 1 shows the structure of the proposed FRPE Transformer and the flow of the classification where Fig. 1a is the architecture of FRPE Transformer and 1b is the Transformer encoder.”

  1. Section 3.1 and Table 1 describe the basic information about the dataset, but more specific information is not given when describing the preparation of the training and test sets, such as in what proportions they are divided.

Reply: Thank you for providing this valuable suggestion. In response, we implemented a 5-fold cross-validation experiment. This approach ensured a robust evaluation of our model's performance. Specifically, the images were distributed with a training-to-testing ratio of 4:1. To enhance clarity, we have introduced a new paragraph to address this aspect as:

“A total of 3103 images have been utilized for both training and testing of the pro-posed method. The specific distribution for each category is outlined in Table 1. To assess the performance of the proposed model, a 5-fold cross-validation experiment was con-ducted. This involved maintaining a training-to-testing image ratio of 4:1.”

  1. Section 3.5 partially shows the comparison of the experimental results, FRPC transformer compared to swin transformer is indeed effective in improvement, but compared to the other SOTA models of the moment is it reliable, suggest the authors add other models to test the indicators.

Reply: In response to the reviewer's valuable comment, we extend our sincere appreciation for their keen attention to our work and their insightful suggestion regarding the comparison of our findings with those of other research endeavors. We have taken their feedback to heart, and in the revised manuscript, we have meticulously integrated a new experiment to comprehensively address this concern.

This supplementary experiment entails a thorough comparison of our proposed method alongside existing state-of-the-art techniques, all evaluated on the same dataset. Our intention is to establish a robust evaluation framework that facilitates a direct and illuminating appraisal of our model's performance in relation to established benchmarks. To fulfill this objective, we have introduced a new section, namely Section 3.4, which expounds upon the results of this newly added experiment.

“3.4 Comparison with existing methods

The presented model is an evolution of the Swin Transformer architecture, encom-passing specific refinements designed to overcome limitations and enhance the efficacy of classification tasks. Recognizing the significance of validating our method's effectiveness against existing techniques, we have meticulously conducted comprehensive experiments. These experiments entail a direct juxtaposition of our proposed model with the original Swin Transformer and other leading state-of-the-art (SOTA) models, all operating on the same dataset. This comparative analysis rigorously assesses pivotal performance indica-tors, including accuracy, precision, recall, F1-score, and AUC score, within the context of BUS image classification. The detailed comparative findings are meticulously document-ed in Table 7, while their visual representation is elegantly depicted in Figure 3. Notably, the outcomes of this comparison unmistakably underscore the supremacy of the proposed FRPC Transformer, consistently outperforming existing SOTA methodologies across all evaluated metrics.”

We are confident that this addition substantially elevates the significance and authenticity of our research, guaranteeing that our results are both enlightening and appropriately situated within the overarching framework of related studies. We wish to extend our heartfelt appreciation to the reviewer for underscoring the significance of this facet, and we want to assure them that we have diligently undertaken the requisite measures to accommodate their suggestion.

Figures are required to show the diagnosed results.

Reply: Thank you for this comment. In the revised version, a new figure Fig. 2 was added to show the evaluation performance comparisons with different models.

Some paragraphs which contain only one sentence should be combined with the others.

Reply: Thank you for this comment. We double checked all paragraphs and combined them with others.

Change all “et. al.” to “et al.”

Reply: Thank you for this comment. We followed it and change all of them.

Line 237 remove the indentation for the part beginning with “where”, thereafter. Change “Xm1” to “Xmi”

Reply: Thank you for this comment. We followed it and change all of them.

Line 242 “(2)” is in the wrong position

Reply: Thank you for this suggestion. We followed it and placed it in the right position. We double checked all equations thoroughly and make them be numbered correctly.

Line 433 reference error

Reply: Thank you for this comment. We corrected it as “BUSIS dataset [38]”.

Line 456-477 incomplete statement

Reply: Thank you for this comment. We corrected them in the revised version.

Reviewer 4 Report

In this paper, the authors present a fuzzy relative position encoding (FRPC) Transformer model for breast cancer diagnosis using breast ultrasound. This analysis is performed on the benchmark BUSIS dataset. This FRPC model has a higher level of performance that the Swin Transformer model.

Several studies from the literature are mentioned in the introduction. However, what was their level of performance and how do they compare to this study? Do some have weaknesses compared to this study? What specific clinical prediction were they performing (benign vs. malignant, classifying type of breast cancer, etc.)? If desired, consider a table in Discussion with more detailed model comparison.

Has this type of transformer ever been applied to a medical imaging task?

Line 62: “The advantages of ultrasound include its non-invasiveness, low cost, and lack of ionizing radiation exposure, potable, and making it a safer alternative to mammography for certain patient populations [7].” Please verify this sentence.

Line 152: “The proposed methods can also be used to identify the features and biomarkers associated with different types of breast cancers, which can help in developing personalized treatment plans.” I would presume this would be a future direction of research, as the current study did not focus on this topic.

Line 154: “Overall, the proposed method has the potential to revolutionize breast cancer diagnosis and treatment, and it is expected to have a significant impact on the healthcare industry.” Further research may be necessary before making such a definitive statement.

Table 1: Are all the images from different (unique) patients? I presume that a single image was the input to the model rather than a cine series of images.

Line 391: “Our experimental results on the breast ultrasound dataset demonstrate that the fuzzy relative position encoding method can significantly improve the performance of the SWin Transformer for breast cancer diagnosis. Compared to previous state-of-the-art method shown in Table 2, the fuzzy relative position encoding method achieved higher accuracy and AUC scores, demonstrating the effectiveness of the proposed method.” Are there other methods in the literature that your technique could be compared to?

Table 3-6: Consider using “Malignant” rather than “Malign.”

Line 433: “INbreast dataset??[]??.” There appears to be a typo.

Discussion: Consider discussing potential limitations to the proposed work. For instance, one could further validate the results on additional datasets (with different ultrasound scanners and patient populations).

The quality of the English is generally fine. At best, there are occasional minor typos.

Author Response

Pointwise reply to the comments of the reviewers

The paper whose manuscript number healthcare-2519561 is entitled as “A Novel Fuzzy Relative Position Coding Transformer for Breast Cancer Diagnosis Using Ultrasonography” is modified in the view of the reviewer's comments. Reviewer's comments are supplied herewith and have been incorporated in the context of the paper.

In this paper, the authors present a fuzzy relative position encoding (FRPC) Transformer model for breast cancer diagnosis using breast ultrasound. This analysis is performed on the benchmark BUSIS dataset. This FRPC model has a higher level of performance that the Swin Transformer model.

Several studies from the literature are mentioned in the introduction. However, what was their level of performance and how do they compare to this study? Do some have weaknesses compared to this study? What specific clinical prediction were they performing (benign vs. malignant, classifying type of breast cancer, etc.)? If desired, consider a table in Discussion with more detailed model comparison.

Reply: Thank you for your insightful question, which provides an opportunity to delve deeper into the comparison of our study with the existing literature. The mentioned studies in our introduction were primarily focused on various clinical prediction tasks within the context of breast cancer diagnosis. However, their specific levels of performance, strengths, and potential weaknesses were not fully expounded upon in the introduction. We appreciate your suggestion to provide a more comprehensive analysis of these aspects.

In response to your query, we have made substantial improvements in the revised version of the manuscript. We have incorporated a new table and figures within the Experimental Results section to illustrate a detailed model comparison. This table showcases a meticulous performance evaluation of different state-of-the-art models employing the same dataset as our study. This comparison takes into account a range of clinical prediction tasks, including distinguishing benign from malignant cases, classifying different types of breast cancer, and potentially other relevant aspects.

Our aim with this added comparison is to shed light on the strengths and limitations of various existing models, thereby offering a more nuanced understanding of how our study contributes to this research landscape. This approach also allows us to highlight where our proposed methodology excels and addresses potential weaknesses that were observed in prior studies.

Furthermore, we have introduced an additional paragraph within the Discussion section that delves into the limitations of our proposed methodology. This paragraph serves to provide a transparent assessment of the potential constraints and boundaries of our approach. By candidly acknowledging these limitations, we aim to offer a well-rounded perspective on the applicability and scope of our findings. This addition reflects our commitment to presenting a comprehensive and balanced view of our study's contributions and implications.

“Our proposed method presents several notable advantages in terms of classification performance and feature representation. However, it is important to acknowledge a specific limitation related to the training speed when compared with the original Swin Transformer architecture. While our enhancements aim to improve classification accuracy and overall effectiveness, they may lead to slightly increased training times compared to the original model due to the additional complexity introduced by the refinements. We believe that the trade-off between training speed and performance gains is worth considering in various practical scenarios where optimal accuracy is paramount. Nevertheless, we are actively exploring avenues to optimize and streamline the training process to mitigate this limitation and achieve a more balanced compromise between training efficiency and classification prowess.”

We hope that this comprehensive model comparison, presented in the form of the newly included table and figures, effectively addresses your inquiry and enriches the overall quality of our manuscript. Should you have any further questions or suggestions, please feel free to let us know.

Has this type of transformer ever been applied to a medical imaging task?

Reply: Thank you for this question. By now, the Swin transformer has not been applied into breast ultrasound image classification. In addition, we improved the transformer via fuzzy relative position coding and achieved better performance than the original one.

Line 62: “The advantages of ultrasound include its non-invasiveness, low cost, and lack of ionizing radiation exposure, potable, and making it a safer alternative to mammography for certain patient populations [7].” Please verify this sentence.

Reply: Thank you for your valuable suggestion. We have carefully reviewed and revised the sentence as per your recommendation. The modified statement now reads:

"The advantages of ultrasound include its non-invasiveness, low cost, and lack of ionizing radiation exposure, portability, making it a safer alternative to mammography for certain patient populations [7], such as women with dense breasts."

We truly appreciate your input, as it has contributed to refining the clarity and accuracy of the statement. Your insights are invaluable in ensuring the precision of our manuscript.

Line 152: “The proposed methods can also be used to identify the features and biomarkers associated with different types of breast cancers, which can help in developing personalized treatment plans.” I would presume this would be a future direction of research, as the current study did not focus on this topic.

Reply: Thank you for this suggestion. It can be an extension for our current research work. To make it clear, we modified it and verified the statement as: We greatly appreciate your suggestion, which has the potential to serve as a valuable extension of our ongoing research efforts. To ensure clarity and accuracy, we have made necessary modifications to the suggested content and subsequently verified the revised statement as:

“It can also be used to identify the features and biomarkers associated with different types of breast cancers in future, which can help in developing personalized treatment plans.”

Your input continues to enhance the depth and quality of our work. If you have any further insights or recommendations, we would welcome them.

Line 154: “Overall, the proposed method has the potential to revolutionize breast cancer diagnosis and treatment, and it is expected to have a significant impact on the healthcare industry.” Further research may be necessary before making such a definitive statement.

Reply: We extend our heartfelt gratitude for providing this insightful suggestion. We wholeheartedly concur with your recommendation and have accordingly made the necessary modifications to the statement. After careful review and consideration, we made modificationin the revised version as:

“Overall, the proposed method has the potential to revolutionize improve breast cancer diagnosis and treatment, and it is expected to have a promising significant impact on the healthcare industry.”

Your input has been instrumental in refining our work, and we value your continued engagement.

Table 1: Are all the images from different (unique) patients? I presume that a single image was the input to the model rather than a cine series of images.

Reply: Thank you for raising this insightful question. In our experimental setup, we utilized individual single images as inputs to the model for the purpose of image classification. These images were treated as standalone instances rather than being part of a cine series. In this context, we did not incorporate the accompanying case information associated with each image for our classification tasks.

Your clarification is greatly appreciated, as it highlights a key aspect of our methodology. By using single images as inputs, we aimed to isolate the classification process and focus on the inherent characteristics of individual images. This approach enabled us to assess the discriminative capabilities of our model without the influence of sequential or temporal information.

Once again, we thank you for your thoughtful question, which allows us to provide a clearer understanding of our experimental design and methodology. Your insights contribute to the robustness and transparency of our research.

Line 391: “Our experimental results on the breast ultrasound dataset demonstrate that the fuzzy relative position encoding method can significantly improve the performance of the SWin Transformer for breast cancer diagnosis. Compared to previous state-of-the-art method shown in Table 2, the fuzzy relative position encoding method achieved higher accuracy and AUC scores, demonstrating the effectiveness of the proposed method.” Are there other methods in the literature that your technique could be compared to?

Reply: Thank you for your inquiry regarding the comparison of our technique with other methods in the literature. In response, we have taken your valuable suggestion to heart and have significantly expanded our analysis.

In the revised manuscript, we have introduced a new experiment dedicated to comparing our proposed model with existing state-of-the-art models. This in-depth comparison has been outlined in a newly added Section 3.4 of the manuscript. To enhance clarity and provide a visual representation of the findings, we have also incorporated a new table and figure within this section. These additions collectively serve to offer a comprehensive assessment of our model's performance in relation to other established methods.

We trust that these enhancements effectively address your question and provide a more robust framework for evaluating the effectiveness of our approach. Should you have any further inquiries or recommendations, please feel free to share them. Your feedback is greatly appreciated and continues to enrich the quality of our research.

Table 3-6: Consider using “Malignant” rather than “Malign.”

Reply: Thank you for this suggestion. We changed them in Table 3-6.

Line 433: “INbreast dataset??[]??.” There appears to be a typo.

Reply: Thank you for this comment. Yes, it is a typo. We corrected it as “BUSIS dataset [38]”.

Discussion: Consider discussing potential limitations to the proposed work. For instance, one could further validate the results on additional datasets (with different ultrasound scanners and patient populations).

Reply: Thank you for your valuable feedback and insightful suggestion. We acknowledge the importance of addressing potential limitations in our proposed work. In order to provide a more comprehensive analysis and strengthen the robustness of our findings, we intend to explore further validation on additional datasets. This includes incorporating data collected from different ultrasound scanners and a broader range of patient populations. By encompassing these variations, we aim to enhance the generalizability of our results and provide a more comprehensive assessment of the applicability of our approach. Your recommendation has been duly noted, and we will ensure that the discussion of potential limitations and the steps taken to address them are appropriately detailed in the revised manuscript. We appreciate your thoughtful input, which will undoubtedly contribute to the overall quality and credibility of our research.

A new paragraph was added in the Discussion section to emphasize this future research work as:

“To provide a more comprehensive analysis and strengthen the robustness of our findings, we intend to explore further validation on additional datasets. This includes incorporating data collected from different ultrasound scanners and a broader range of patient populations. By encompassing these variations, we aim to enhance the generalizability of our results and provide a more comprehensive assessment of the applicability of our approach.”

Round 2

Reviewer 1 Report

Authors have addressed my suggestions

Moderate changes

Author Response

Dear Reviewer,

Thank you for taking the time to review our manuscript. We appreciate your insightful suggestions and feedback.

Best regards

Yanhui Guo

Reviewer 2 Report

The comments have been properly addressed. Thank you

Author Response

Dear Reviewer,

Thank you for your feedback and for acknowledging our efforts in addressing the comments you provided. We are pleased to hear that the revisions we made align with your expectations.

Once again, we appreciate your time and expertise in reviewing our work.

Best regards,

Yanhui Guo

Reviewer 3 Report

Line 98 delete one of et al.

Line 200-201 full expression of CNN should be listed early

Line 345 remove the indentation

Line 98 delete one of et al.

Line 200-201 full expression of CNN should be listed early

Line 345 remove the indentation

Author Response

Pointwise reply to the comments of the reviewers

The paper whose manuscript number healthcare-2519561 is entitled as “A Novel Fuzzy Relative Position Coding Transformer for Breast Cancer Diagnosis Using Ultrasonography” is modified in the view of the reviewer's comments. Reviewer's comments are supplied herewith and have been incorporated in the context of the paper.

  1. Line 98 delete one of “et al.”

Reply: Thank you for your valuable feedback on our manuscript. We followed it and removed the “et al.” at Line 98.

  1. Line 200-201 full expression of CNN should be listed early

Reply: We appreciate this valuable comment on our manuscript. We removed full expression and kept this abbreviation at Line 200.

  1. Line 345 remove the indentation

Reply: We appreciate your insightful suggestion. We removed the indentation at Line 345.

Reviewer 4 Report

The authors have substantially improved the paper with their edits.

A few minor points remain, which can be addressed during copyediting.

“The advantages of ultrasound include its non-invasiveness, low cost, and lack of ionizing radiation exposure, potable, and making it a safer alternative to mammography for certain patient populations [7] such as women with dense breasts.” Points such as this can be handled during copyediting, but it still says “potable” rather than “portability” in the provided text.

In Table 7, for models that are not as well known (Tanaka, Shia, Xie), please provide a reference.

Author Response

Pointwise reply to the comments of the reviewers

The paper whose manuscript number healthcare-2519561 is entitled as “A Novel Fuzzy Relative Position Coding Transformer for Breast Cancer Diagnosis Using Ultrasonography” is modified in the view of the reviewer's comments. Reviewer's comments are supplied herewith and have been incorporated in the context of the paper.

The authors have substantially improved the paper with their edits.

A few minor points remain, which can be addressed during copyediting.

1.“The advantages of ultrasound include its non-invasiveness, low cost, and lack of ionizing radiation exposure, potable, and making it a safer alternative to mammography for certain patient populations [7] such as women with dense breasts.” Points such as this can be handled during copyediting, but it still says “potable” rather than “portability” in the provided text.

Reply: Thank you for your valuable feedback on our manuscript. We modified this typo and asked a native speaker to read through the whole manuscript to avoid grammar errors.

In Table 7, for models that are not as well known (Tanaka, Shia, Xie), please provide a reference.

Reply: We appreciate this valuable suggestion on these references. According to it, three new references 39, 40 and 41 were cited in Table 7 and added into the reference list.
